# Bovine Fibroblast-Derived Extracellular Matrix Promotes the Growth and Preserves the Stemness of Bovine Stromal Cells during In Vitro Expansion

**DOI:** 10.3390/jfb14040218

**Published:** 2023-04-13

**Authors:** Kathleen Lee, Anisha Jackson, Nikita John, Ryan Zhang, Derya Ozhava, Mohit Bhatia, Yong Mao

**Affiliations:** 1Laboratory for Biomaterials Research, Department of Chemistry and Chemical Biology, Rutgers University, 145 Bevier Rd., Piscataway, NJ 08854, USA; kwl49@scarletmail.rutgers.edu (K.L.); atj43@scarletmail.rutgers.edu (A.J.); nikita.john@rutgers.edu (N.J.); ryanzhang01@gmail.com (R.Z.); do311@chem.rutgers.edu (D.O.); 2Atelier Meats, 666 Burrard Street, Suite 500, Vancouver, BC V6C 3P6, Canada

**Keywords:** cultivated meat, extracellular matrix, stem cells, stemness, adipogenic differentiation, cell expansion

## Abstract

Cultivated meat is a fast-growing research field and an industry with great potential to overcome the limitations of traditional meat production. Cultivated meat utilizes cell culture and tissue engineering technologies to culture a vast number of cells in vitro and grow/assemble them into structures mimicking the muscle tissues of livestock animals. Stem cells with self-renewal and lineage-specific differentiation abilities have been considered one of the key cell sources for cultivated meats. However, the extensive in vitro culturing/expansion of stem cells results in a reduction in their abilities to proliferate and differentiate. Extracellular matrix (ECM) has been used as a culturing substrate to support cell expansion for cell-based therapies in regenerative medicine due to its resemblance to the native microenvironment of cells. In this study, the effect of the ECM on the expansion of bovine umbilical cord stromal cells (BUSC) in vitro was evaluated and characterized. BUSCs with multi-lineage differentiation potentials were isolated from bovine placental tissue. Decellularized ECM prepared from a confluent monolayer of bovine fibroblasts (BF) is free of cellular components but contains major ECM proteins such as fibronectin and type I collagen and ECM-associated growth factors. Expansion of BUSC on ECM for three passages (around three weeks) resulted in about 500-fold amplification, while cells were amplified less than 10-fold when cultured on standard tissue culture plates (TCP). Moreover, the presence of ECM reduced the requirement for serum in the culture medium. Importantly, the cells amplified on ECM retained their differentiation abilities better than cells cultured on TCP. The results of our study support the notion that monolayer cell-derived ECM may be a strategy to expand bovine cells in vitro effectively and efficiently.

## 1. Introduction

Conventional meat production is currently facing challenges stemming from environmental and animal welfare concerns [1,2,3]. Conventional meat production uses excessive amounts of land and freshwater and emits a substantial amount of greenhouse gases. This has limited its further expansion to meet the growing demands for meat, which are estimated to double between 2017 and 2050 [1,4]. It is inevitable to explore alternatives. Cultivated meat, which shares similarities at the cellular level with conventional meat, is produced by utilizing many technologies developed and optimized for tissue engineering and regenerative medicine [1,5,6]. To make cultivated meat, two components are needed: various types of cells and scaffolds. The two key types of cells are muscle and fat cells, which make up >90% of the mass of meat [7]. While progenitor cells can differentiate into myocytes or adipocytes, in vitro culture and expansion (to increase the number) of progenitor cells are limited [8,9]. A large quantity of cells is needed to make cultivated meat. According to an estimate, 1 kg of wet meat alone requires roughly 2.9 × 10^11^ muscle cells [10]. To this end, a sustainable cell source is required.

Many cell sources have been explored, including muscle satellite cells, adipose-derived stem cells, multipotent mesenchymal stem cells, induced pluripotent stem cells, transgenics, and immortalized cells [2]. Among them, multipotent mesenchymal stem cells (stem cells) stand out as promising candidates due to their abilities of self-renewal, differentiation, and their relatively easy acquirement from multiple types of tissues [1,11]. Stem cells or progenitor cells used in cultivated meat in current literature are mostly isolated from tissue biopsies via enzymatic digestion. While biopsies are small injuries to animals, to avoid harming them at all, other cell sources should be considered. For example, stem cells isolated from birth tissues such as the placenta, which are often discarded as waste, can be a potentially sustainable cell source. Stem cells derived from placental tissues, including the amniotic and chorionic membranes, umbilical cord tissues, and amniotic fluid, showed multipotent characteristics [12].

In vitro expansion of cells is a critical step to achieving a significant number of cells for not only cultivated meat but also for tissue engineering and cell-based therapy. Many insights gained from the in vitro expansion of human cells, including human stem cells, are valuable guides in developing strategies for the effective expansion of animal stem cells. Long-term culture of human stem cells results in decreased proliferation and loss of stemness [13,14,15,16,17,18]. For example, the growth and stemness of human bone marrow-derived mesenchymal stem cells significantly diminished within a few passages under conventional cell culture conditions [17]. In vitro expansion of stem cells isolated from human umbilical cord tissue results in the premature senescence of cells [13]. In order to overcome this limitation, many approaches have been taken [2]. Growth factors such as the fibroblast growth factor and the hepatocyte growth factor have been used in cell culture to promote the growth of stem cells and to maintain the stemness of cells during in vitro cell culture [19]. In vitro culture/expansion of stem cells under standard culture conditions cause stem cells to adhere to and grow on a cell culture-treated plastic surface. This environment is far from the in vivo microenvironment, where the cells naturally reside. The expansion of stem cells in this environment can cause a changed cell shape and an altered expression of genes and proteins [20]. Therefore, to mimic the natural microenvironment, extracellular matrix (ECM) derived from tissues or ECM produced and assembled by cells cultured in vitro have been explored to support the growth of stem cells and primary cells [17,21,22,23]. Among these studies, ECM prepared from bone marrow-derived stem cells has shown significant positive effects on the expansion of human stem cells [17,23]. Since growing stem cells to generate stem cell ECM may be costly, an alternative cell source for making ECM should be explored.

Fibroblasts are the predominant cell type in the interstitial tissue matrix and are the main cell type to produce and assembles ECM [24,25]. In vitro culture of fibroblasts has been robust and effective since the emergence of cell culture technology one hundred years ago [26,27]. Fibroblasts showed phenotypic and cell surface similarity to stem cells. This suggests that the ECM produced by fibroblasts and stem cells may share significant similarities [28]. Mitotically inactivated fibroblasts are used as feeder cells for embryonic stem cells and induced pluripotent stem cells. This indicates the importance of fibroblasts for providing the ECM and growth factor components needed for the growth of stem cells [29]. While fibroblast feeder cells cannot be used as off-the-shelf cell culture substrates, the ECM derived from fibroblasts may have the potential to be an off-the-shelf cell culture substrate used to support the in vitro expansion of stem cells.

To explore this possibility, this study investigated the in vitro expansion of bovine stem cells isolated from umbilical cord tissues. Bovine fibroblast-derived ECM, hereafter referred to as BF-ECM, was prepared by decellularization of monolayer culture of bovine fibroblasts. The expansion of bovine stem cells on BF-ECM and standard cell culture substrates was compared over multiple passages. The stemness of expanded bovine stem cells was evaluated through their potential to differentiate into an adipogenic lineage after multiple passages. Culturing bovine stem cells on BF-ECM significantly enhanced the expansion efficiency compared with culturing cells under the standard condition (tissue culture treated polystyrene, TCP). More importantly, after expansion for many passages, cells (even at passage 9) expanded on BF-ECM and retained the adipogenic differentiation ability significantly better than cells expanded on TCP. Additionally, the presence of ECM reduced the requirement for serum in the culture medium and provided a venue to explore the cost-effectiveness of cell expansion. Furthermore, the presence of growth factors such as platelet-derived growth factor (PDGF) and transforming growth factor beta 1 (TGF-β1) in BF-ECM was detected by ELISA assays. This suggests that sequestering growth factors by ECM may be one of the explanations for the positive effect of BF-ECM on bovine stem cells.

In summary, our study is the first report on utilizing a robust cell-derived ECM as a culturing substrate that not only promotes the proliferation of bovine stem cells but also retains their stemness in later passages, providing a means to achieve a high quantity and high quality of animal cells for cultivated meat technology.

## 2. Materials and Methods

### 2.1. Isolation and Culturing of Bovine Fibroblasts and Bovine Umbilical Cord Stem Cells

The bovine fibroblasts were isolated from the skin of a 1-month-old male calf by enzymatic digestion as described previously [22] The bovine fibroblasts (BF) were cultured in a DMEM medium (Gibco, Waltham, MA, USA) containing 10% fetal bovine serum (FBS). Bovine umbilical cord stem cells were isolated from cow placental tissues following published methods with modifications [30,31,32]. It has been determined that this research does not require Rutgers University’s institutional animal care and use committee (IACUC) approval since there is no intervention with or interaction with animals. The tissue is normal, natural birth tissue otherwise disposed of as waste in the farming facility. The tissue handling and cell isolation from such tissue have been approved by the Rutgers Environmental Health & Safety (REHS) Biosafety Protocol (Code #15-081). Briefly, bovine placental tissues were collected after a birth at the New Jersey Agricultural Experiment Station (Rutgers University, New Brunswick, NJ, USA), and the umbilical cord tissues were harvested and cleaned in a biosafety cabinet. Cord tissues were digested in digestion solution (MEM, a complete medium + 1 × Antibiotic − Antimycotic + 1 mg/mL of collagenase type I (Worthington Biochemical Corporation, Lakewood, NJ, USA) at 5 mL/gram of tissue. After digestion in the tissue culture incubator with rocking for 3 h, an equal volume of PBS was added to the digestion mixture and then passed through cell strainers (45 μm pore). The fraction passing through the strainer is referred to as “cell fraction” and the undigested tissues are referred to as the “tissue fraction”. “Tissue fraction” was then washed using PBS twice and cultured (about 5 segments per dish) in 10 cm cell culture dishes (Cat #229621, CELLTREAT, Pepperell, MA, USA) in MEM-alpha medium with 10% FBS and 1 × Antibiotic-Antimycotic (anti/anti) (ThermoFisher, Branchburg, NJ, USA) for 7–10 days. Cells were harvested for subculture or cryopreservation as bovine umbilical cord stem cells (BUSC) P1 cells.

### 2.2. Evaluation of the Multipotency of BUSC

To evaluate the multipotency of isolated BUSCs, 2 × 10^4^/well of BUSCs were seeded to 24-well TCP plates (Cat #229124, CELLTREAT, Pepperell, MA, USA). After incubation for 2 days, cells were induced using adipogenic medium (DMEM medium containing 10% FBS and 7 different fatty acids: myristoleic acid, pristanic acid, phytanic acid, erucic acid, elaidic acid, oleic acid, and palmitoleic acid at 50 μM) [25,33] or osteogenic medium (α-MEM medium containing 10% FBS and supplemented with 20 mM β-glycerophosphate, 50 µg/mL ascorbic acid, and 100 nM dexamethasone) [34] following the published protocols. After adipogenic induction for 6 days, cells were stained with Oil Red O following published protocols [35,36]. After osteogenic induction for 11 days, cells were fixed and stained with 0.4 mL/well of Fast Blue RR solution (1/10 of a capsule of Fast Blue RR salt (Sigma FBS25) in 4.8 mL of H_2_O and 200 μL of Naphthol AS-Mx (Sigma) at room temperature for 30 min [37]. After staining, cells were analyzed using an ECHO microscope (BICO, Boston, MA, USA).

### 2.3. Preparation and Characterization of BF-ECM

Bovine fibroblasts were seeded in 6-well tissue culture treated plates (Cat #229124, CELLTREAT, Pepperell, MA, USA) and cultured in DMEM medium + 10% FBS with 100 µM of ascorbic acid/well until reaching 100% confluence. The decellularization of the monolayer culture was carried out following a procedure modified from a published protocol [38]. Briefly, once cells reached confluence, the medium was removed from each well. Each well was then washed with 1 mL/well of PBS twice, followed by the decellularization with 1 mL/well of 0.5% Triton X-100 with 20 mM NH_4_OH. Plates were incubated at room temperature for 5 min. After incubation, the decellularization solution was removed, and each well was gently washed with 1 mL/well of dH_2_O. The washed ECM was left to air dry in the tissue culture hood. After drying, the ECM was washed with 1 mL/well of dH_2_O an additional three times. The washed ECM was used immediately or air-dried in the biosafety cabinet and stored at 4 °C until use.

The prepared BF-ECM was characterized by the quantification of the residual DNA, SDS-PAGE, and immunofluorescent staining using antibodies against type I collagen or fibronectin. All characterization procedures followed the published methods and are described briefly below [22,39,40].

### 2.4. Immunofluorescent Staining of BF-ECM

To detect the presence of proteins in the BF-ECM or cells, immunofluorescent staining was performed, as previously described [41]. Briefly, BF-ECM or cells (without decellularization) were fixed with 4% paraformaldehyde for 1 h and permeabilized in 0.5% Triton X100 in PBS for 1 h. The fixed and permeabilized samples were stained with primary antibodies, which were diluted in staining buffer (1 × PBS + 5% FBS + 0.02% NaN3), and the staining was incubated at 4 °C overnight. The primary antibodies used in this study are anti-bovine fibronectin polyclonal antibody (Millipore Sigma AB2047, Burlington, MA, USA) at 1:40; anti-bovine type I collagen polyclonal antibody (Millipore Sigma AB749P) at 1:40. After the incubation with primary antibodies, samples were washed and stained with secondary antibodies, goat anti-rabbit IgG-Alexa 555 antibody (Ref #A21428, Life Technology) or goat anti-mouse IgG-Alexa 488 (Ref#A21042, Life Technology, Carlsbad, CA, USA) at 1:500 for 1 h. Nuclei were stained with Hoechst dye 33258, 20 mM in water (Cat #83219, AnaSpec Inc., Fremont, CA, USA) at 1:500 for 5–15 min. Staining samples were imaged under an epi-fluorescent microscope (Zeiss Axio Observer D1, Jena, Germany).

### 2.5. Quantification of Residual DNA in BF-ECM

The DNA in confluent cell cultures of bovine fibroblasts before and after decellularization was quantified as follows. Cells or ECM were extracted with an extraction buffer containing 1% SDS and 1% Triton X-100. The DNA in the extractants was isolated by using Phenol/Chloroform/Isoamyl alcohol and precipitated by ethanol. The isolated DNA was then quantified using the Helixyte Green dsDNA Assay Kit (AAT Bioquest, Pleasanton, CA, USA).

### 2.6. In Vitro Expansion of BUSCs on BF-ECM and Tissue Culture Treated Polystyrene (TCP)

The amplification of cells over multiple passages was monitored as described [21]. Briefly, 5 × 10^4^/well of BUSC (P2) were seeded onto BF-ECM in 6-well plates or onto TCP in 6-well plates. After culturing for 7–10 days, when cells reached about 80–90% confluence, cells from two wells (one column of the six wells) were trypsinized, combined, and counted using hemocytometers as one sample. The fold of amplification was calculated as the number of cells at harvest divided by the number of cells at seeding. The means of the folds of amplification from three samples (totaling six wells) were reported as mean ± SD. 2–10 × 10^4^/well P3 cells were seeded and cultured. These cell amplification steps were repeated for multiple passages. The overall fold amplification was calculated by multiplying the folds of amplification of multiple passages [21]. The doubling time at each passage was calculated as follows [42]: $Doubling\;time\;\left(h\right)=\frac{\left(Duration\;\left(h\right)\times \ln2\right)}{\ln\frac{Final\;Concentration}{Intital\;Concentration}}$.

### 2.7. Monitoring of Cell Adhesion and Proliferation

The adhesion and growth of cells (BUSCs expanded on BF-ECM or TCP) in the presence of different percentages of bovine fetal serum (FBS) were monitored as described [39]. Briefly, 2 × 10^4^ cells were added per well (2.6 mg/cm^2^) of a 48-well TCP. Plates were incubated at 37 °C with 5% CO_2_ and 95% humidity. After incubation for 24 h (Day 1), the medium was removed. The adhered cells were detected using the alamarBlue assay (a metabolic activity assay). Briefly, 0.2 mL/well of alamarBlue solution (complete growth medium + 10% alamarBlue reagent) (Bio-Rad Laboratories, Philadelphia, PA, USA) was added to each well and incubated at 37 °C for 30 min. After incubation, 0.1 mL/well of supernatant was transferred to a 96-well plate. Fluorescent intensity was read using a multimode microplate reader (Spark^®^, TECAN, Mannedorf, Switzerland) at excitation/emission (Ex/Em) equal to 540 nm/590 nm. Fluorescent intensity was expressed in arbitrary units (AU).

To monitor the growth of these adhered cells, after the alamarBlue assay, cells were washed with 0.5 mL/well of PBS once. 0.2 mL/well of the medium was added to each well and cultured in the tissue culture incubator for 1–2 h. After the recovery from the alamarBlue assay, the medium was removed and 0.2 mL/well of fresh medium was added to each well, which was then continued to culture to the next time point. At each time point, the cell viability was measured using the alamarBlue assay. The percentage of growth at each time point was determined as: FL_(day T)/_FL_(day 1)_ × 100%.

### 2.8. Adipogenic Differentiation of BUSCs

Cells expanded on BF-ECM or expanded on TCP were differentiated on ECM or TCP; cells expanded on BF-ECM or TCP was cultured to 75% confluence, trypsinized, and counted. 10 × 10^4^/well of cells were seeded into the wells of 6-well plates with BF-ECM or TCP, respectively. After overnight incubation, cells were induced to undergo adipogenic differentiation using the Stempro Adipogenesis Differentiation Kit (ThermoFisher Scientific). The medium was changed every three days for 18 days. After differentiation, cells were stained with Oil Red O following a published protocol [36].

Cells expanded on BF-ECM or expanded on TCP were differentiated on TCP: after expanding cells on BF-ECM or TCP for multiple passages, ECM cells or TCP cells were seeded to 24-well TCP plates at 2 × 10^4^/well. After culturing the cells for 2 days, an adipogenic medium made of seven different fatty acids was added [33]. After induction for 7 days, quantitative Oil Red O staining of cells was performed following a published protocol [35]. Briefly, cells were fixed using 4% paraformaldehyde and rinsed with phosphate-buffered saline (PBS). Fixed cells were stained with 500 μL of the Oil Red O working solution (0.18% Oil Red O in 60% isopropanol) for 15 min at room temperature. After staining, wells were rinsed using 300 μL distilled water four 4 times. The dye was eluted by the addition of 500 μL of isopropanol per well and incubated on a rocker or plate shaker for 15 min. The absorbance of eluents was detected at 490 nm/570 nm using a TECAN-plated reader (Spark^®^, TECAN, Mannedorf, Switzerland). To normalize the Oil Red O staining to the number of cells under different conditions, a separate set of cells was cultured and treated in parallel, then lysed using cell lysis buffer (Cell Signaling Technology, Beverly, MA, USA), and the DNA of each sample was determined using the Helixyte Green assay. The Oil Red O staining of each sample was expressed as the OD490 nm/μg DNA.

### 2.9. Determination of Growth Factors in BF-ECM Using ELISA Assays

To determine if growth factors are present in the decellularized BF-ECM, BF-ECM or BF cultures without decellularization (BFs) were extracted with 1.2 mL/well/sample extraction buffer (2 M urea, 2 mM phenylmethylsulfonyl fluoride (PMSF), 0.1 M K-PBS). After extraction at room temperature with vigorous shaking for 6 h, the extractants were collected. To remove urea, samples were dialyzed in a Slide-A-Lyzer Dialysis Cassette–3500 MWCO (Thermo Scientific) in 3 L of PBS for 48 h, with one change of PBS after 24 h at 4 °C. The presence of fibroblast growth factor 2 (FGF2), PDGF subunit A (PDGFA), and transforming growth factor-beta 1 (TGF-β1) in the extractants were quantified using ELISA kits from Abclonal Technology (Woburn, MA, USA). ELISA assays (FGF2 Cat# RK09056, PDGFA Cat# RK06228, TGF-β1Cat# RK04050) were performed following the manufacturer’s instructions.

### 2.10. Statistical Analysis

Statistical analysis was performed as described [41]. Each independent experiment contained 3 or more biological repeat samples (*n* ≥ 3), and data are presented as the mean ± standard deviation. The results shown are representative of at least two independent experiments. A one-way ANOVA with a Tukey’s multiple comparisons test was performed to determine statistical significance using GraphPad Prism, GraphPad Software (La Jolla, CA, USA, https://www.graphpad.com, version 9.4.1 for Mac OS X, access date: 18 July 2022) for all quantitative data except for Table 1. The statistical differences between ECM and TCP at different passages were determined using a student Excel *t*-test. Differences were considered significant at a *p*-value of < 0.05.

## 3. Results and Discussion

### 3.1. Isolation and Characterization of Bovine Umbilical Cord Stem Cells (BUSC)

To avoid harvesting cells from live animals via tissue biopsies, stromal cells from otherwise discarded birth tissue are used in this study. Stromal cells isolated from bovine umbilical cord tissues have been shown to be multipotent and can be differentiated into multiple lineages [30,31]. To verify if BUSCs isolated for this study are multipotent, cells were cultured in MEM-alpha growth medium and induced to differentiate using adipogenic medium or osteogenic medium (Figure 1). Cells showed positive adipogenic differentiation as indicated by oil red O staining (Figure 1B) and positive osteogenic differentiation as indicated by fast blue staining (Figure 1D). These results indicate that the isolated BUSCs have multi-lineage differentiation ability. A set of stem cell markers has been chosen for characterizing human mesenchymal stromal cells [43]. A preliminary flow cytometry analysis was performed on BUSCs and showed that these cells are CD45^−^ CD29^+^ CD44^+^ (Appendix A).

### 3.2. Preparation and Characterization of Decellularized ECM from BFs

A decellularized ECM has been defined as a matrix that contains the desirable ECM components and is free of cellular components [44]. In this study, an ECM was prepared from a confluent culture of bovine fibroblasts by a decellularization process described in Methods (Figure 2A). The presence of key ECM proteins (fibronectin and type I collagen) in decellularized ECM was verified by immunofluorescent staining (Figure 2B). Interestingly, an elaborate fibrillar matrix became more prominent after decellularization.

This was likely due to masking by the intracellular staining of type I collagen.

The residual DNA in the decellularized BF-ECM was extracted and quantified in comparison with the BF culture before decellularization (Figure 2C). These results demonstrated that the decellularization thoroughly removed DNA (an indicator of cellular components) while retaining fibronectin and type I collagen in their fibrillar matrix form in BF-ECM.

### 3.3. The Growth of BUSC on BF-ECM and TCP

To evaluate if different culturing substrates affect the growth of BUSCs, cells were cultured on BF-ECM and TCP in a MEM-alpha growth medium for multiple passages. Cells on ECM looked denser and adopted more elongated morphologies compared with cells on TCP (Figure 3).

The growth of BUSC at different passages (P3 and P5) on BF-ECM versus TCP was monitored using a viability assay (alamarBlue assay) (Figure 4). Cells in passage 3 showed better growth on BF-ECM than on TCP (Figure 4A). The number of cells was trypsinized and counted on Day 9 (Figure 4B). This count confirmed that there were more cells on the BF-ECM than on the TCP after culturing for the same amount of time. A similar experiment was performed on cells at passage 5 (Figure 4C). Consistent with cells at P3, cells grew faster on BF-ECM than on TCP. These results indicate that BF-ECM has a positive effect on the growth of BUSCs, and such effects persist for multiple passages.

### 3.4. Expansion of BUSC on BF-ECM and TCP

To monitor the cell expansion on BF-ECM and TCP, cells at passage 3 (P3) were seeded onto BF-ECM or TCP at the same densities and cultured for 7–10 days. After incubation, cells were trypsinized (P4), counted, and set up in new cultures on BF-ECM or TCP. This cell expansion process was repeated for three passages. Two batches of BUSCs that were isolated from umbilical cord tissues from two different cows were tested (Table 1).

Batch 1 BUSCs showed 539× amplification from P3 to P6 on BF-ECM in 24 days, while only 5.2× amplification was achieved on TCP (Table 1). Batch 2 BUSCs amplified 486× from P3 to P6 on BF-ECM in 21 days. In comparison, cells were amplified by about 8.9× on TCP in the same amount of time. These results suggested that BF-ECM is a better culture substrate to support cell expansion in vitro. This is consistent with the previous reports that ECM derived from primary cells promotes in vitro cell expansion [21,23]. Two different batches of BUSCs isolated from different umbilical cord tissues showed similar patterns of in vitro expansion. This not only strengthened the observation but also suggested a consistency between different batches of cells.

### 3.5. The Presence of BF-ECM Reduced the Requirement of FBS in Cell Culture Medium

Cells were cultured in MEM-alpha complete medium containing 10% fetal bovine serum (FBS). Since the serum is the most expensive component in the cell culture medium, it would be cost-effective to grow cells in the reduced serum. To test if the presence of ECM may reduce the requirement of serum for growth, cells were cultured in complete medium (with 10% FBS) or with reduced serum (2% or 5% FBS) on BF-ECM or TCP for 11 days (Figure 5).

As shown in Figure 5A, cells on BF-ECM grew as well, if not better, in 2% FBS or 5% FBS than cells in 10% FBS on TCP for up to 7 days (Figure 5B). Cells reached near confluency on BF-ECM with 5% or 2% FBS by Day 7. After 7 days, cells continued to grow in 10% FBS until confluence, when the viability of cells in 2% FBS started to decrease, likely due to insufficient nutrients. On the other hand, cells on TCP showed FBS concentration-dependent growth. Even with 10% FBS, the growth of cells on TCP is significantly less than that of cells on BF-ECM with 2% FBS. This observation suggested that when cells are cultured on ECM, the requirement for serum may be reduced.

It is interesting to note that 10% FBS delayed the growth of cells on BF-ECM in comparison with media with lower concentrations of FBS. Even though the specific cause is not known, this phenomenon has been reported previously [45]. The reduced requirement of FBS to achieve the same or even better growth of stem cells can be explored further for the cost-effectiveness of in vitro cell culture.

### 3.6. The Adipogenic Differentiation Potential of Busc Was Preserved Better in Cells Expanded on BF-ECM than on TCP

To evaluate the stemness of in vitro expanded cells, the adipogenic differentiation ability of expanded cells was monitored. The adipogenic differentiation of BUSCs was initially performed using StemPro adipogenic medium, which is an optimized adipogenic medium for human mesenchymal stem cells (Appendix A). After 18 days, the percentage of Oil Red O-positive stained cells over the total number of cells was higher with cells on BF-ECM than with cells on TCP (Appendix A). However, the overall adipogenic differentiation of BUSC occurred at a very low frequency. This result indicated that the StemPro adipogenic kit may not be an effective induction medium for these bovine cells.

An adipogenic formula was optimized by modifying a fatty acid-rich adipogenic medium [32]. To evaluate the adipogenic differentiation potential of cells expanded on BF-ECM and TCP, cells expanded on BF-ECM or TCP for 8 passages were seeded onto 24-well TCP plates. Cells were induced with the optimal adipogenic induction medium (Figure 6). After induction for 7 days, cells were stained with Oil Red O (Figure 6A) and quantified by dye extraction and normalization to the number of cells (DNA) (Figure 6B).

It was noticed that the number of cells expanding on TCP decreased over time during induction, and only some cells produced oil droplets. On the other hand, the oil droplet formation is prominent, with cells expanded on the BF-ECM. The oil red O was extracted from the stained cells and quantified. To normalize the staining with the number of cells, a separate set of cells, which were cultured and treated in parallel, were lysed. The DNA of each cell lysate was quantified using the Helixyte Green assay. As shown in Figure 6B, the Oil Red O staining was significantly higher in cells expanded on BF-ECM. These results indicate that cells expanded on BF-ECM retained their potential to be differentiated into an adipogenic lineage even after long-term expansion.

### 3.7. The Presence of Growth Factors in BF-ECM

The mechanism by which the ECM supports cell growth and phenotype maintenance is complex. This includes directly enhancing the interaction with cells via binding to integrins, 3D architecture, and mechanical properties that mimic the cell’s natural microenvironment [21,46,47,48]. ECM components bind to growth factors and increase the local concentrations of such growth factors, which directly regulate the cellular functions of cells residing within [49,50]. Many growth factors, including fibroblast growth factor 2 (FGF2), platelet-derived growth factor (PDGF), and transforming growth factor-beta 1 (TGF-β1), have been shown to be associated with the ECM via ECM proteins or heparan sulfate proteoglycans (HSPG) [51]. FGF2 binds to heparan sulfate with high affinity and promotes cell growth [52,53]. PDGF is associated with the ECM via HSPG and direct interaction with fibronectin and plays important roles in the proliferation and migration of cells [54,55]. TGF-β1 is a pleiotropic growth factor that regulates cellular functions such as proliferation, differentiation, and motility [56]. The activation and sequestration of TGF-β1 are closely mediated by the ECM [53,57,58]. To explore if these growth factors are associated with BF-ECM, decellularized BF-ECM (ECM) and BF monolayer confluent culture (ECM/cells) were extracted with an extraction buffer containing 2 M urea. Then the presence of FGF2, PDGFA, or TGF-β1 in extracts was quantified using ELISA assays (Figure 7).

While the levels of FGF2 in the extracts were below the assay detection limits, the presence of TGF-β1 or PDGF was detected in the BF-ECM and ECM/cells. There was no significant difference between growth factors extracted from BF-ECM and from ECM/cells, which suggested that these growth factors were mostly associated with BF-ECM. Sequestering growth factors by BF-ECM may improve the accessibility of growth factors to cells. It may also be a mechanism of action in promoting the growth and preserving the stemness of BUSCs. Furthermore, it is known that stem cells produce abundant growth factors [59]. It is likely that the ability of the ECM to sequester growth factors would confer even more positive effects during the culturing/expansion of stem cells.

While the sequestering of growth factors may be one of the mechanisms ECM employs to positively regulate cellular functions, the composition and mechanical properties of ECM most likely play important roles in supporting cell growth and phenotype maintenance by mimicking cells’ natural environment. The use of bovine fibroblast-derived ECM as a culture substrate to amplify bovine stem cells has not been reported. The practicality of using BF-ECM as culture substrates for cultivated meat requires additional studies on the efficacy, cost-effectiveness, and scalability of this ECM in comparison with other approaches such as cocktails of purified growth factors [19].

## 4. Conclusions

In vitro expansion of bovine stem cells is an important step for cultivated meat. To overcome the negative effects of long-term in vitro culturing on the proliferation and stemness of stem cells, a bovine fibroblast-derived ECM (BF-ECM) was prepared and utilized as a culturing substrate for in vitro expansion of bovine umbilical cord stem cells. Our results can be summarized as follows: bovine fibroblast-derived ECM (BF-ECM)

-retains the key proteins fibronectin and type I collagen after decellularization;-is a promising cell culture substrate to support bovine stem cell expansion in vitro faster and better;-reduces the requirement for FBS in the cell culture medium;-retains the stemness of bovine stem cells at later passages.

Since BF-ECM can easily be prepared as off-the-shelf culture substrates, it may be a promising approach to achieving the high quantity and quality of stem cells required for cultivated meat.

## Figures and Tables

**Figure 1 jfb-14-00218-f001:**
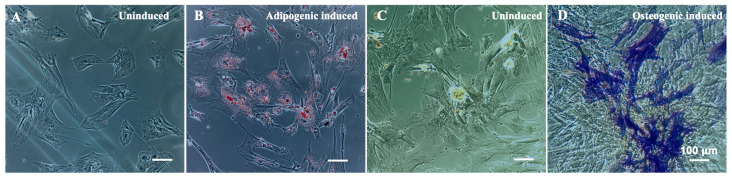
The multipotency of isolated BUSCs. BUSCs were cultured in growth medium (**A**) or adipogenic medium (**B**) for 6 days and stained with Oil Red O. BUSCs were cultured in growth medium (**C**) or osteogenic medium (**D**) for 11 days and stained with Fast Blue. Representative images were shown. Scale bar = 100 μm.

**Figure 2 jfb-14-00218-f002:**
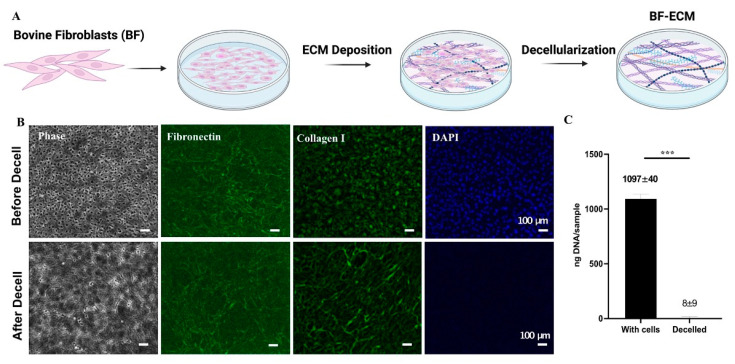
Preparation and characterization of BF-ECM. (**A**) The preparation of BF-ECM is illustrated (**A**). Phase contrast and fluorescent images of a BF confluent culture (upper panel, Before Decell) and BF-ECM (lower panel, After Decell) (**B**). The staining of antibodies and DNA dye was indicated in the images. Scale bar = 100 μm. The quantification of residual DNA in monolayer BFs and decellularized BF-ECM (**C**). Data shown are mean ± SD (*n* = 4) *** *p* < 0.005.

**Figure 3 jfb-14-00218-f003:**
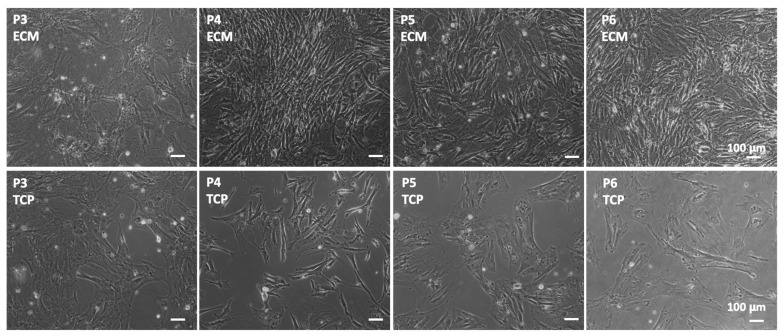
BUSCs cultured on BF-ECM and TCP for multiple passages. BUSCs at different passages were seeded at 5 × 10^4^/well on BF-ECM (upper panel) or TCP (lower panel) 6-well plates. After being cultured for 7 days, phase contrast images were taken. Representative images were shown. Scale bar = 100 μm.

**Figure 4 jfb-14-00218-f004:**
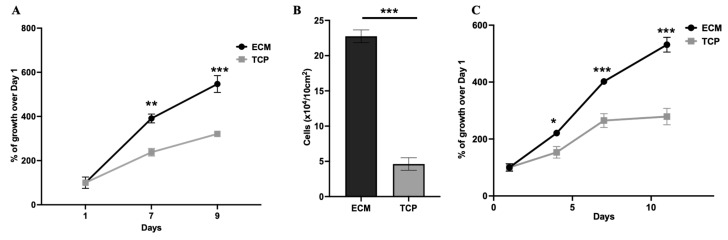
The growth of BUSCs at different passages on BF-ECM versus TCP. The cell viability of BUSCs at P3 cultured on BF-ECM (ECM) or TCP was monitored for 9 days. The % of viability at each time point was normalized to the viability on Day 1. % of growth = viability_(Day N)_/viability_(Day 1)_ × 100% (**A**). The P3 cells on BF-ECM or TCP were collected and counted on Day 9 (**B**). The viability of cells at P5 was monitored for 11 days. Data shown are mean ± SD (*n* = 4) (**C**). Statistical differences between ECM and TCP are indicated for each time point * *p* < 0.05, ** *p* < 0.01 *** *p* < 0.001.

**Figure 5 jfb-14-00218-f005:**
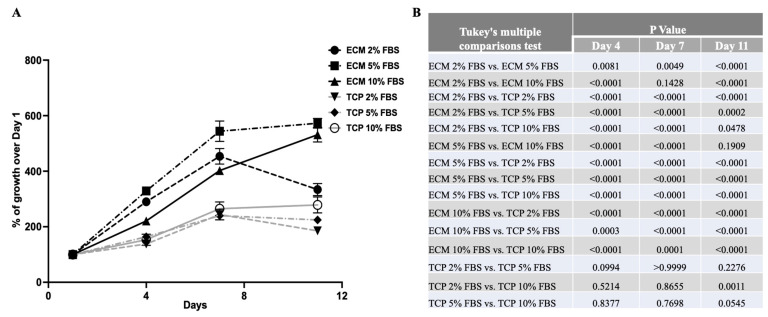
The growth of BUSCs (P3) on BF-ECM (ECM) versus TCP in the presence of different percentages of bovine serum. (**A**) The cell viability was measured using the alamarBlue assay. % of growth = viability_(Day N)_/viability_(Day 1)_ × 100%. Data shown are mean ± SD (*n* = 3). (**B**) The statistical differences (*p* values) between different culturing conditions at each time point were determined using one-way ANOVA with Tukey’s multiple comparisons tests (GraphPad Prism).

**Figure 6 jfb-14-00218-f006:**
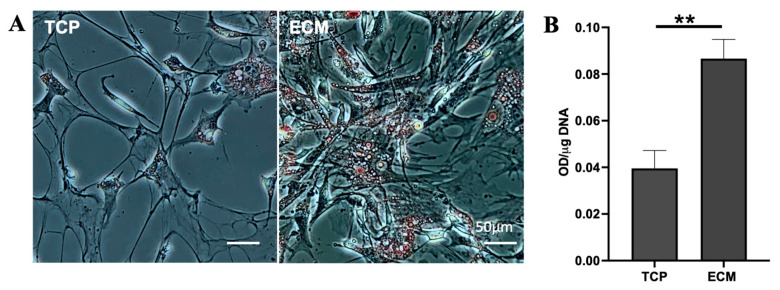
Adipogenic differentiation of BUSCs expanded on TCP or BF-ECM with an optimized adipogenic medium. BUSCs were expanded on TCP or BF-ECM (ECM) till passage 8. Cells (at P9) were seeded on TCP, induced with an optimized adipogenic medium for 7 days, and stained with Oil Red O (**A**) Scale bar = 50 μm. (**B**) Oil red O dye was extracted from stained cells and quantified and normalized to the number of cells (μg of DNA). Data shown are mean ± SD (*n* = 3) ***p* < 0.01.

**Figure 7 jfb-14-00218-f007:**
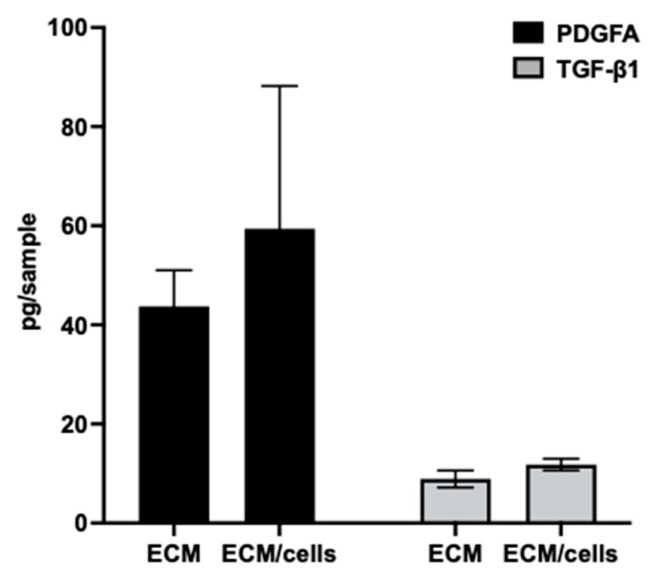
Determination of growth factors in BF-ECM. BF-ECM (ECM) and BF monolayer confluent culture (ECM/cells) were extracted with 2 M urea, and the presence of growth factors in extracts was determined using ELISA assays. Data shown are mean ± SD (*n* = 3).

**Table 1 jfb-14-00218-t001:** The expansion of two batches of BUSC on BF-ECM or TCP for three passages. Batch 1 BUSCs were expanded from P3 to P6 on BF-ECM (ECM) or TCP for 24 days. The fold of amplification and doubling time were calculated for each passage (6 wells of BF-ECM and 6 wells of TCP) as described in Materials and Methods. The mean ± SD of three samples for each condition is shown. The statistical difference (*p*-value) between ECM and TCP for each passage was determined by a student *t*-test. The total theoretical expansion (fold) from P3 to P6 was calculated by multiplying the folds of expansion of all 3 passages. Batch 2 BUSCs were expanded on BF-ECM or TCP similarly for 21 days.

**Batch 1**	**P3** **→** **P4**	**P4** **→** **P5**	**P5** **→** **P6**	**Total**
Amplification (fold) on ECM	2.2 ± 0.09	25 ± 0.3	9.6 ± 1.3	539
Doubling time (h) on ECM	185	41	51	
Amplification (fold) on TCP	0.6 ± 0.06	4.7 ± 0.1	1.9 ± 0.5	5.2
Doubling time (h) on TCP	N/A	86	179	
Statistical difference (*p* value)	0.005	0.0017	0.0006	
**Batch 2**	**P3** **→** **P4**	**P4** **→** **P5**	**P5** **→** **P6**	**Total**
Amplification (fold) on ECM	6.4 ± 0.6	9.8 ± 0.3	7.8 ± 0.7	486
Doubling time (h) on ECM	81	58	57	
Amplification (fold) on TCP	2.5 ± 0.15	2.8 ± 0.8	1.3 ± 0.4	8.9
Doubling time (h) on TCP	162	129	485	
Statistical difference (*p* value)	0.0043	0.0048	0.0006	

## Data Availability

Data available on request due to restrictions of a research agreement.

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
