# Peer review of "Bovine Fibroblast-Derived Extracellular Matrix Promotes the Growth and Preserves the Stemness of Bovine Stromal Cells during In Vitro Expansion"

_jfb, 2023, doi:10.3390/jfb14040218_

Round 1

Reviewer 1 Report

Summary

This manuscript reported the culture system of bovine umbilical cord stromal cells (BUSCs) on decellularized extracellular matrix (ECM) of bovine fibroblasts. ECM containing fibronectin, collagen I, platelet-derived growth factor (PDGF), and transforming growth factor (TGF) enhances proliferation and maintains adipogenic differentiation ability of BUSCs. Stem cell culture of ECM will be a useful technology for cultivated meat production in addition to regenerative medicine.

Most of the experiments appear to have been conducted appropriately, but some claims should be qualified along with the results. Some additional description and statistical analysis is also required. Discussion is not sufficient to appeal the novelty of this study. The manuscript needs to be moderately improved for publication in Journal of Functional Biomaterials.

Comments

1.          “bovine umbilical cord stromal cells” should be abbreviated to “BUSCs”. In some sentences of the manuscript, the authors described “BUSC cells”, but this is not appropriate because “BUSC” already contains “cells” as defined. Similarly, “bovine fibroblasts” can be abbreviated to “BFs”, and “BF cells” is not appropriate because “fibroblasts” already means “cells”.

2.          Materials and Methods: The bovine fibroblasts and umbilical cord stem cells were isolated from animals. The authors must describe the Use of Animals as required by the Journal Instructions (https://www.mdpi.com/journal/jfb/instructions). Especially, I was interested in the information (age, sex, etc.) of the animals that provided fibroblasts, because the ECM components may differ among individuals.

3.          Figures 4A and 4C: The statistical differences between ECM and TCP need to be analyzed and displayed.

4.          Table 1: Were these experiments performed only one time (single well) per batch? If the experiments were performed multiple times, the value should be presented with SD, and the statistical differences between EMC and TCP need to be analyzed and displayed. The number of experiments (wells) should be noted in the text or legend.

5.          Figure 5: The statistical differences among the groups should be analyzed and displayed.

6.          Figure 5: The growth rate of BUSCs are not different between 2% and 10% FBS even on TCP. This result means that 2% FBS is sufficient to culture BUSCs either on ECM and TCP. The authors claimed “BF-ECM reduced the requirement of FBS”, but it is not supported by the data. Did the authors test 0% FBS? I think this condition may show critical difference between ECM and TCP.

7.          Page 9: The authors subtitled “The stemness of BUSC was preserved better in cells expanded on BF-ECM than on TCP”. But the experiments shown in Figures 6 and 7 tested only adipogenic differentiation ability, which is a part of stemness. The claim must be limited as at the end of the first paragraph on Page 11.

8.          I think Figure 6 is better to be prepared as Supplementary Data, because it is negative but immediately overcome by Figure 7. It may be confusing for the readers when their first reading.

9.          The main text consists of “Results and Discussion”, but there is little discussion of the study. Following points should be discussed more. The mechanism by which ECM maintains or enhances cell ability; the advantages of this technology for meat production; the scientific novelty of this study.

Minor points

10.          Abstract: “TCP (tissue culture plate)” should be “tissue culture plate (TCP)”.

11.          Legend of Table 1 and main text (the end of Page 8): “One batch” should be “Batch 1”, and “Another batch” should be “Batch 2”.

Author Response

Please see the attachment, thanks.

Reviewer 2 Report

1.      The term “Stemness” is misleading/unscientific.

2.      In this study, the effect of ECM on the expansion of bovine umbilical cord stromal cells (BUSC)………”, the term “expansion” is also unscientific. In the abstract, the term “expansion” and “expand” for in vitro bovine umbilical cord stromal cells is seems misleading. Tissue expansion would still make sense. Cells either proliferate or differentiate.

3.      Authors have frequently used the term “culture” alone, which is not appropriate. It should be “tissue culture/ cell culture.”

4.      The authors claim “In vitro culture/expansion of stem cells under standard culture condition subject stem cells to adhere and grow on a cell culture treated plastic surface” is misleading. The cells grow on plastic surface that is treated with collagen, and collagen is one of the major components of extracellular matrix. Primary difference being that of monolayer and multilayer wherein cell proliferation and differentiation significantly varies.

5.      In the phrase, “(tissue culture treated polystyrene, TCP) “, The tissue culture flask or plates are itself is made of polystyrene material, and then surface treated/coated with collagen, otherwise adherent-type cells would not attach and proliferate on the surface. It is important that the authors mention the brand and catalogue number of the tissue culture plates used as well.

6.      The authors have claimed that fibroblast derived ECM could be a better replacement of serum for cell culturing based on the presence of few growth factors, however, we do not accept this as ECM is used as a scaffold rather than supplement. Serum contains far more variety of proteins and growth factors that gives the cells a wholistic supply of supplements. Principally, when comparison is done between ECM and serum, then ECM should also be dissolved in the medium like serum to have unbiased comparison that ECM is an alternative to serum. Indeed, the number of cells is increasing in one batch but that has got more to do with presence or absence of 3D ECM scaffold that provides greater and multidimensional surface area for anchorage of adherent cells as compared to monolayer of collagen coat present in TCP.

7.      In the phrase, “robust cell-derived ECM as a culturing substrate”, ECM is not a substrate.

8.      We do not see an unbiased experimental setup wherein BUSC grown in 3D-ECM scaffold is compared to BUSC grown as a monolayer in TCP (probably coated with single layer of collagen).

9.      Materials used require more detailing.

Author Response

Please see the attachment, thanks.

Reviewer 3 Report

Dear Authors,

It is a big focus on the topic, but I would like to tell you that this document needs modifications before being considered, see the following list of points that need to be considered:

- The introduction is very crowded with a lot of information included making it difficult to read. I would suggest the authors reduce the size making it lighter and easier to understand.

- I suggest including among the references the work of Dominici et al. on the identification criteria of MSCs. “Minimum criteria for the definition of multipotent mesenchymal stromal cells. Position Statement of the International Society for Cellular Therapy” which reads as follows: “MSCs should be plastic adherent when maintained under standard culture conditions. Second, MSCs must express CD105, CD73 and CD90 and lack expression of CD45, CD34, CD14 or CD11b, CD79alpha or CD19 and HLA-DR surface molecules”. Sorry to be harsh but the guidelines say so. But being non-human but bovine cells at least the following CD90 CD105 CD44 positive markers and CD14CD34 negative producers as B.Rossi et al. "Isolation and in vitro characterization of stem cells derived from bovine amniotic fluid in different trimesters of pregnancy", and a representative figure with the histogram overlays of the markers with the respective isotype controls.

- The guidelines of Dominici et al. indicate to perform the triple differentiation chondrogenic, fat and bone, then we must add the chondrogenic differentiation.

- I suggest inserting a "Flow cytometry" section in materials and methods and inserting in this paragraph how the staining was performed, number of washes, RPM and centrifugation time, make and model of the flow cytometer, name and version of the analysis software. Also, I would list or create a table with marker, isotype, clone, dilution, manufacturer.

  - In the results and discussion section, there are many long sentences that make the reader lose count. I would suggest the author to design a scheme that organizes and sorts his comments on the results in a simpler and more understandable way.

My best regards

Author Response

Please see the attachment, thanks.

Round 2

Reviewer 1 Report

The authors sincerely revised the manuscript according to the reviewers' comments.

Reviewer 2 Report

The manuscript has been significantly revised. The authors have answered all the major queries satisfactorily. 

Reviewer 3 Report

Dear authors,

Now it's ok for me, with the suggested corrections

and additions.

My best regards.